# Profiling transcription factor activity dynamics using intronic reads in time-series transcriptome data

**Yan Wu**[1,2,3,4☯], **Lingfeng Xue**[1,2,3☯], **Wen Huang**[1,2,3☯], **Minghua Deng**[1,4], **Yihan Lin**[1,2,3]*

1 Center for Quantitative Biology, Academy for Advanced Interdisciplinary Studies, Peking University, Beijing, China, 2 The MOE Key Laboratory of Cell Proliferation and Differentiation, School of Life Sciences, Peking University, Beijing, China, 3 Peking-Tsinghua Center for Life Sciences, Academy for Advanced Interdisciplinary Studies, Peking University, Beijing, China, 4 School of Mathematical Sciences and Center for Statistical Science, Peking University, Beijing, China

☯ These authors contributed equally to this work.
* yihan.lin@pku.edu.cn

**Data Availability Statement:** RNA sequencing data of Jurkat T cells have been deposited to GEO and can be accessed from: https://www.ncbi.nlm.nih. gov/geo/query/acc.cgi?acc=GSE178827. The

## Abstract

Activities of transcription factors (TFs) are temporally modulated to regulate dynamic cellular processes, including development, homeostasis, and disease. Recent developments of bioinformatic tools have enabled the analysis of TF activities using transcriptome data. However, because these methods typically use exon-based target expression levels, the estimated TF activities have limited temporal accuracy. To address this, we proposed a TF activity measure based on intron-level information in time-series RNA-seq data, and implemented it to decode the temporal control of TF activities during dynamic processes. We showed that TF activities inferred from intronic reads can better recapitulate instantaneous TF activities compared to the exon-based measure. By analyzing public and our own time-series transcriptome data, we found that intron-based TF activities improve the characterization of temporal phasing of cycling TFs during circadian rhythm, and facilitate the discovery of two temporally opposing TF modules during T cell activation. Collectively, we anticipate that the proposed approach would be broadly applicable for decoding global transcriptional architecture during dynamic processes.

## Author summary

Many health-related cellular processes, such as immune response and disease progression, involve dynamic changes of gene expression state, which are orchestrated by transcription factors. Dissecting the activities of transcription factors is thus important for understanding cellular processes and for interfering with dysregulated processes. Our ability to analyze transcription factor activities has been facilitated by genome-wide gene expression data from high-throughput assays such as RNA sequencing. Existing methods typically estimate transcription factor activities based on the expression levels of matured mRNAs of target genes. However, because the levels of matured mRNAs are affected by

codes used for calculations and producing figures in the paper can be accessed from: https://github.com/TFactivity/TFA. Information regarding the public datasets used in this manuscript is included in S1 Table. Source data for figures can be found in S2 Table.

**Funding:** This study was supported by funding from National Key R&D Program of China, under grant numbers 2020YFA0906900 (YL) and 2018YFA0900703 (YL), and from National Natural Science Foundation of China under grant numbers 31771425 (YL) and 32088101 (YL). The funders had no role in study design, data collection and analysis, decision to publish, or preparation of the manuscript.

**Competing interests:** The authors have declared that no competing interests exist.

transcriptional and post-transcriptional regulatory activities, the estimated transcription factor activities may not faithfully recapitulate the regulatory activities of transcription factors. In this paper, we proposed and validated an alternative approach for analyzing transcription factor activities using the expression levels of unmatured mRNAs of target genes, allowing us to decode how transcription factor activities are temporally controlled during key biological processes. Our results provide insights into the temporal phasing of key circadian regulator activities in mouse liver, and uncover two temporally opposing modules of transcription factors that dictate the immune responses in T cells. Therefore, this approach can help understand the regulatory principles of dynamic cellular processes.

This is a *PLOS Computational Biology* Methods paper.

## Introduction

Cells control the activities of transcription factors (TFs) to orchestrate temporal gene expression programs during diverse cellular processes such as stress response and cell differentiation [1, 2]. During these processes, TF activities can be controlled at the level of protein abundance, post-translational modification, and/or subcellular localization [3], leading to changes in transcription rates of downstream target genes. Due to diverse control mechanisms, TF activity dynamics can exhibit different timescales [4–6], ranging from minute-level nuclear translocation pulses of stress response TFs [7–9], to the daily activity rhythm of circadian TFs [10]. However, it has been challenging to decode the TF activity dynamics with high temporal accuracy.

To potentially overcome the preceding challenge, we first discussed the existing methods for characterizing TF activity dynamics. A key approach is to track TF activities in single cells using time-lapse imaging by monitoring the TF's nuclear expression level, nuclear localization, or the expression of target reporters [11]. This approach facilitated the discovery of TFs with complex activity dynamics such as p53 [12, 13] and NF-κB [14, 15]. On the other hand, TF activity dynamics can also be quantified by measuring the expression levels of endogenous target genes in time-series snapshots of cells taken over a time course [16]. In this scenario, cells are assumed to be synchronized and TF activity dynamics are measured in different cell populations along the time course. While the former approach measures the same single cells over time and can thus provide a time-lapse measurement of the TF activity dynamics, it is technically challenging to simultaneously study many TFs using this approach.

The latter approach has been greatly facilitated by the availability of time-series genome-wide gene expression data, including microarray and RNA-seq data. Building upon these data and gene regulatory network (GRN), bioinformatic algorithms and tools have made it possible to analyze TF activities globally [17–25]. Generally, these approaches assume that gene expression is the ensemble of TF activities (e.g., network component analysis [17]), or that TF activities are reflected by the ensemble of target expression within each regulon (e.g., AUCell [22] and VIPER [23]). Global TF activity profiles of each sample or cell are then computed through optimization or statistical methods. Importantly, the accuracy is largely affected by the choice of GRN and could be very low when using inferred and non-curated GRN [24, 26]. Meanwhile, TF activities estimated using literature-curated GRN show a higher accuracy compared to the estimates using GRNs inferred from ChIP-seq, transcription factor binding site, or

inference methods [24, 26]. However, TF activities estimated by different methods have relatively low overlaps [27], indicating potential shortcomings of the underlying rationales. Thus, additional improvements are necessary in order to better analyze TF activities.

We next discussed potential ways to improve TF activity estimation using transcriptome data. A key issue of the preceding methods relates to the use of exon-level information in the RNA-seq data, which represents matured mRNAs. This leads to at least two shortcomings of the resulting TF activities: a) they cannot resolve instantaneous TF activities because matured mRNAs represent the temporal integration of the gene's transcriptional activity; b) they capture both transcriptional and post-transcriptional effects and thus may not reflect the actual TF activities alone. To address this, experimental approaches involving nucleotide analog labeling have been established to measure newly synthesized mRNAs [28–30], allowing an accurate genome-wide quantification of transcriptional activities and thus a much-improved estimation of TF activities [31]. While such experimental approaches can address existing shortcomings, it would be desirable to make use of existing sequencing data without the need to acquiring data with new experimental protocols.

Here, we leveraged intron-level information in existing RNA-seq data, and tested an alternative approach for analyzing TF activities using public data as well as newly collected data. The rationale for focusing on intronic sequencing reads is that introns are mostly from unspliced mRNAs, and intronic counts can directly capture the effect of transcriptional regulation [32]. Intron-level information has been utilized in previous studies to better capture transcriptional activities of stress response or circadian genes [33, 34], and to facilitate the inference of differentiation trajectories using single-cell RNA-seq data [35, 36]. We first carried out computational simulations to demonstrate the advantage of using unspliced mRNA counts for TF activity estimation. Next, by using existing RNA-seq data and literature-curated GRN [24], we showed that intron-based TF activities display higher correlations with TF nuclear localization levels and TF chromatin occupancies compared to exon-based TF activities. To decode how TF activities are temporally controlled during key biological processes, we analyzed public circadian rhythm data of mouse liver and collected our own data on T cell response. With these datasets, we provided insights into the temporal phasing of key circadian regulator activities, and uncovered two temporally opposing modules of TFs that orchestrate T cell activation. Together, the proposed approach improves the estimation of instantaneous TF activities and should allow broadly decoding global TF dynamics during various cellular processes using existing datasets and conventional RNA-seq protocols.

## Results

### Model simulations illustrate the advantage of using unspliced target mRNA levels for TF activity estimation

We first used p53 as a generic example for illustrating the potential advantage of using unspliced target mRNA levels for TF activity estimation compared to using mature mRNA levels. p53, a mammalian tumor suppressor, is one of the notable examples of TF with complex temporal dynamics [12, 13]. It has been established that p53 can be activated in temporally discrete pulses during the response to DNA damage stress, and external inputs can modulate the temporal activity dynamics of p53 to control cell fate [12, 13]. Using p53 as a generic example, we depicted two pulses of TF activity, followed by the transcription of a downstream target gene and the splicing of pre-mRNAs (**Fig 1A**).

In the cartoon, unspliced pre-mRNA level follows TF activity closely because the half-life of pre-mRNA is generally short due to the fast rate of splicing [37]. In contrast, matured mRNA level accumulates due to the generally longer half-life of matured mRNA compared to pre-

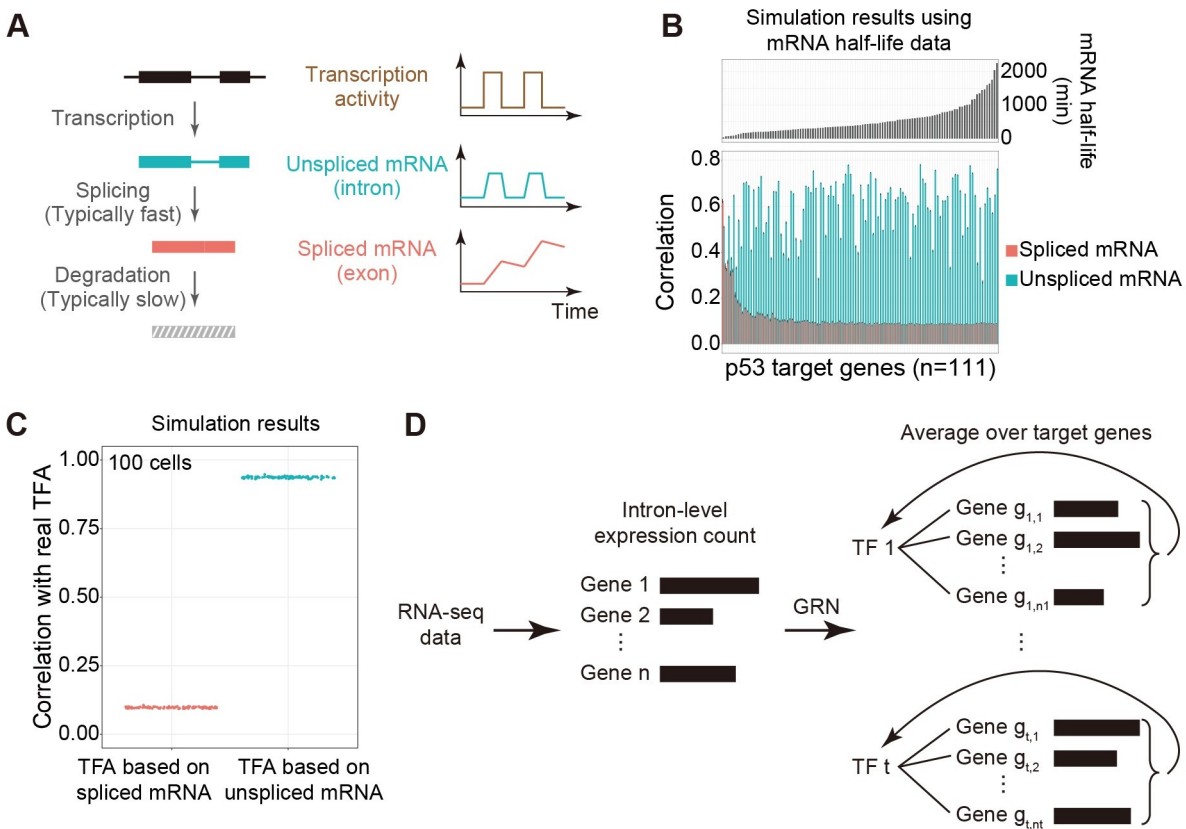

**Fig 1. Simulation-based comparison of using spliced and unspliced mRNA levels of target genes for TF activity estimation. (A)**
Schematics of RNA transcription, splicing, and degradation (left), and the dynamics of unspliced and spliced mRNAs in the presence of
dynamically switching transcriptional activity (right). The unspliced mRNA level can generally better follow the activity dynamics compared
to the spliced mRNA level (right). **(B)** Pearson correlations between simulated p53 activity dynamics and simulated unspliced or spliced
mRNA levels of target genes (bottom) of different mRNA half-lives (top). Correlations were calculated using stochastically simulated single-
cell trajectories as in **S1A Fig**. Unspliced, but not spliced, mRNA levels show consistently high correlations with p53 activity irrespective of
mRNA half-life. Error bars indicate standard errors of 100 simulated cells and n indicates the number of p53 target genes. **(C)** Correlations
between input p53 activities in the simulation and p53 activities estimated using simulated spliced or unspliced mRNA levels. Estimated p53
activities were calculated by averaging the mRNA levels of target genes in the p53 regulon (as in **B**). TFA: Transcription factor activity. **(D)** A
pipeline for the estimation of instantaneous TF activity using intron-level information in the RNA-seq data and curated TF regulon
information. Normalized intron read counts of target genes in the regulon are averaged as the estimate of intron-based TF activity. As a
comparison, TF activity has been conventionally estimated using exon-level information.

mRNA. Under such a depicted scenario, it is apparent that the pre-mRNA level should provide
a better estimate for TF activity compared to matured mRNA level. However, it is also appar-
ent that such a conclusion depends on the choice of half-life parameters.

To systematically analyze the advantage of using unspliced pre-mRNA level for estimating
TF activity, we performed in silico simulations of the p53 system and compared the responses
of different p53 target genes (with different mRNA stability parameters) to oscillatory p53
activity signals (**Figs S1A and 1B**). For each target gene, we chose the TF binding parameter
from a pre-defined range and set the mRNA half-life as in the literature (see **Materials and
Methods**). We found that the correlation between target gene's matured mRNA level and the
input TF activity decreases as mRNA half-life increases (**Fig 1B**), consistent with previous
studies [38, 39]. In contrast, the correlation between unspliced pre-mRNA level and TF activity
remains mostly invariant with respect to mRNA half-life, and is typically much higher than the
preceding correlation. Note that the fluctuation in the correlation coefficient reflects the

difference in the TF binding affinity, which was randomly chosen for each gene from a pre-defined range (**Materials and Methods**).

These results demonstrate the advantage of using unspliced pre-mRNA level compared to matured mRNA level for analyzing TF activity under most scenarios especially when mRNA stability is high.

## Estimating TF activities using unspliced mRNA levels of targets in the regulon

We next explored whether unspliced mRNA levels in a regulon could allow accurate estimation of TF activities. For previous algorithms that use exon-level information, TF activities are estimated by the ensemble of target genes' matured mRNAs within each regulon [22–24]. Compared to these approaches, we reasoned that the ensemble of target genes' pre-mRNAs should allow a more accurate estimation of TF activities based on the preceding results.

By using the simulated responses of p53 target genes in the regulon, we compared the TF activities estimated using the unspliced pre-mRNA levels versus the spliced mRNA levels in the regulon (**S1B Fig**). We simulated 100 cells and found that the estimated TF activities from these two inputs are distinct from each other, with the one estimated using unspliced pre-mRNAs exhibiting a much higher correlation (i.e., much closer to 1) with the input TF activities (**Fig 1C**). Note that the relatively low accuracy of the estimates using matured mRNAs is due to the presence of many highly stable p53 target mRNAs, indicating that the difference in the accuracies between these two estimates is dictated by the overall mRNA stability of target genes in a regulon.

We next investigated the effect of mRNA detection rate in the estimation of TF activities. The rationale is that in single-cell RNA-seq experiments, mRNAs in single cells are often captured at a relatively low rate, e.g., 10%, contributing to sparsity in the expression data. To evaluate this issue, we simulated a specific p53 target gene under varying detection rates (from 5% to 30%) for both unspliced and spliced mRNAs, and computed the correlation between gene expression and input TF activities (**S1C and S1D Fig** and **Materials and Methods**). We found that the low detection rate lowers the correlation for both unspliced and spliced mRNAs (**S1C Fig**). While the advantage of using unspliced pre-mRNA level for TF activity estimation decreases as detection rate reduces, unspliced pre-mRNA generally outperforms spliced mRNA (**S1D Fig**).

More generally, these results indicate that TF activities estimated using unspliced pre-mRNA levels could capture the instantaneous activities of the TF, allowing us to accurately decode the regulatory principles of temporal biological processes.

## Intron-based TF activity displays a higher correlation with TF nuclear localization level than exon-based TF activity

Having established the general advantage of using unspliced pre-mRNA levels of target genes in a regulon using simulated data, we speculated that unspliced pre-mRNA levels approximated by the intronic read counts in the RNA-seq data [32], together with literature curated GRN (that provides regulon information) [24, 40], should allow an accurate estimation of TF activity (**Fig 1D**). To test this, we used two public datasets that measured both transcriptome and TF activity surrogate, allowing us to explore whether intron-based TF activity exhibits a higher correlation with TF activity surrogate compared to exon-based TF activity.

We first used a dataset that measured transcriptome responses proceeding the measurement of TF nuclear localization dynamics in response to stress [41] (**Fig 2A**). More specifically, this dataset focused on a key immune-related TF, NF-κB, which is known to exhibit complex

temporal activity dynamics by translocating between cytoplasm and nucleus [14, 15]. By combining single-cell movie and single-cell RNA-seq [41], this dataset not only provides the individual cells' transcriptome but also the TF nuclear localization dynamics prior to sequencing, allowing us to investigate whether intron-based TF activity could better capture TF nuclear localization level compared to the exon-based measure.

We first compared the population-averaged nuclear localization dynamics of NF-κB (**S2A Fig**) with the population-averaged intron- or exon-based TF activity dynamics. More specifically, we resorted to the time-lapse microscopy data of the nuclear localization dynamics of NF-κB subunit p65 and the single-cell RNA-seq data collected at four different time points during the time course of imaging, and computed averages of the nuclear localization dynamics and the estimated TF activities across single cells at each time point. We found that population-averaged intron-based estimation of NF-κB activity dynamics accurately captures the rise-and-fall of the mean NF-κB nuclear localization dynamics (**Fig 2B** first and second panels). In contrast, exon-based NF-κB activity displays a monotonically increasing trend (**Fig 2B** third panel), and similar results were obtained when using the sum of intronic and exonic reads for activity estimation (**Fig 2B** fourth panel). And as a control, the expression level of the TF itself (i.e. p65) is largely constant (**Fig 2B** bottom). Thus, intron-based TF activity recapitulates the instantaneous NF-κB activity better than exon-based or total read counts-based measure.

We next computed the correlation between NF-κB nuclear localization level and the intron- or exon-based TF activity at the single-cell level for each time point (**Fig 2C**). For the first set of cells that were sequenced 75 min post lipopolysaccharide (LPS) treatment, we found that compared to exon-based measure, intron-based TF activity exhibits a slightly higher correlation with the NF-κB nuclear localization level at the time of cell collection (**Fig 2C** left). As the LPS treatment time increases to 150 min, while both correlations decrease, intron-based TF activity can still better capture the TF nuclear localization level (**Fig 2C** middle). The decrease in correlation as treatment time increases suggests that the target response is more synchronized during the early phase of the stress administration compared to later phases, consistent with the picture that gene activation becomes more stochastic as the TF activity (i.e., NF-κB localization level) approaches a stationary state (**S2A Fig**). At a much later time point post stress (i.e., 300 min), TF nuclear localization no longer shows a significant correlation with target gene activation (**Fig 2C** right).

In these results, because intron-based measures perform only slightly better than exon-based measures, we wondered if such differences are robust to the choice of methods for computing TF activities. We thus resorted to two additional algorithms, VIPER [23] and AUCell [22], which are widely used for computing regulon-based activity scores from gene expression profiles. While these algorithms typically take in exon-level expression profiles, here we used these algorithms to compute TF activities using either exon-level or intron-level expression profiles (**Materials and Methods**). The computed activities were then used to calculate correlations with nuclear localization levels of NF-κB. By doing so, we found that intron-based measures consistently outperform exon-based measures (**Fig 2C**, bottom two rows).

We next investigated whether the improvement of intron-based method over exon-based method is dependent on mRNA stability, as our simulations demonstrated (**Fig 1B**). To do so, we first examined how individual genes' (experimental) mRNA stabilities would affect the relative performance of using intron level versus exon level to report NF-κB nuclear localization at 75 min post LPS stimulation (see **Materials and Methods**). Reassuringly, we found that at the individual gene level, mRNA half-life significantly correlates with the relative performance of intron versus exon (**S2B and S2C Fig**). We next generated two sub-regulons of NF-κB based on the mRNA stability (**S2D Fig**), using which we could separately estimate TF activities and

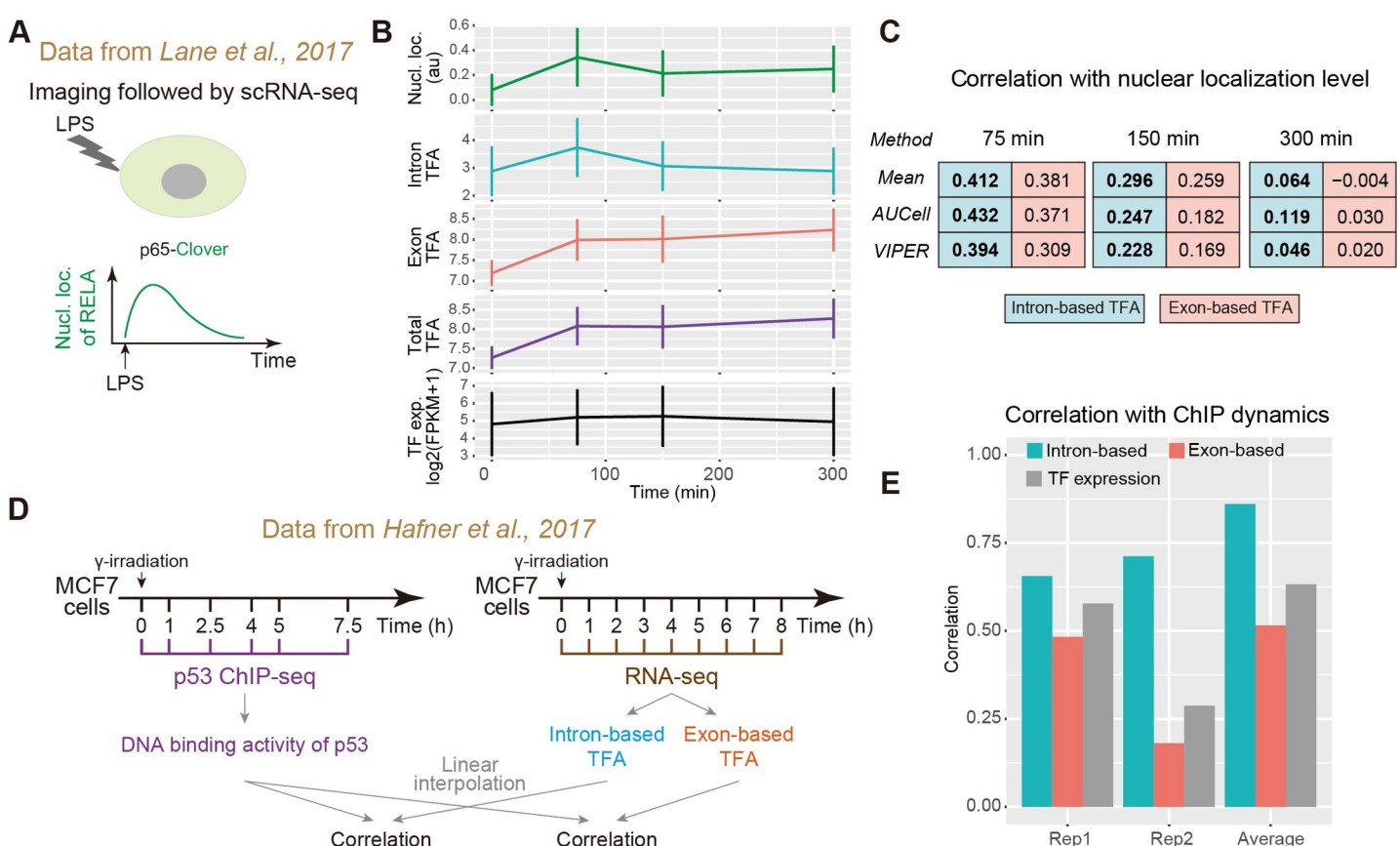

**Fig 2. Validation of intron-based TF activity estimation using public multimodal datasets. (A)** Schematic of the experimental design of Lane et al [41]. After LPS stimulation, the nuclear localization dynamics of NF-κB were recorded in single cells, whose transcriptomes were then sequenced at 75 min, 150 min, or 300 min post LPS stimulation. **(B)** Characterizations of NF-κB activity dynamics using NF-κB nuclear localization level, intron-based NF-κB activity, exon-based NF-κB activity, total reads-based (i.e., intron plus exon) NF-κB activity and NF-κB expression level. Error bars represent standard deviations. Data were taken or reanalyzed from Lane et al [41]. n = 186, 145, 383, and 124 cells from left to right. **(C)** Correlations between single-cell NF-κB nuclear localization level and intron- or exon-based TFA at three time points. For each time point, TFA was calculated by mean expression (as in **Fig 1D**), AUCell [22] or VIPER [23] (see Materials and Methods). **(D)** Schematic of the experimental design of Hafner et al [39] and the steps involved in our analysis. After stimulation by irradiation, bulk p53 ChIP-seq and RNA-seq were performed in different time points. To compute correlation, time series data were linearly interpolated. **(E)** Correlations between p53 DNA binding dynamics computed from ChIP-seq data and intron-based TFA, exon-based TFA, or p53 expression level. 'Rep1' and 'Rep2' represent using data from two separate RNA-seq replicates, and 'Average' represents using the mean of the two replicates.

compare with nuclear localization levels (**S2E and S2F Fig**). We found that intron-based TF activity greatly outperforms exon-based measure for the sub-regulon with long mRNA half-life, but not for the other sub-regulon with short mRNA half-life (**S2F Fig**). Importantly, these results provided strong support for our earlier in silico finding that high mRNA stability limits the accuracy of exon-based TF activity estimation, whereby intron-based method can greatly outperform exon-based method.

Together, these results demonstrate that the nuclear localization-dependent TF activity dynamics are better captured by intron-based TF activity estimates, and that the degree of correlation between the actual TF activity and the estimated TF activity reflects the target activation capacity of the TF, which appears to depend on the stage of the stress response. Moreover, we provided both in silico and experimental evidence that mRNA stability is a key factor affecting TF activity estimation.

## Intron-based TF activity displays a higher correlation with the TF DNA binding activity than exon-based TF activity

We next used a second multi-modal dataset to further evaluate intron-based TF activity estimation. This dataset simultaneously measured transcriptome responses as well as the chromatin occupancy dynamics of p53 in response to DNA damage [39]. More specifically, in this dataset, p53 was activated dynamically in an oscillatory pattern, which was captured by the oscillatory ChIP-seq signals at p53 binding sites, and at the meantime, bulk RNA-seq was performed [39] (**Fig 2D**). This dataset thus allowed us to analyze the DNA binding activity of p53 as well as target genes' gene expression responses at the intronic or exonic level.

Since it is reasonable to assume that the chromatin occupancy of p53 can directly reflect p53 activity, we asked whether intron-based or exon-based TF activity can better correlate with p53 chromatin binding activity. By using a curated list of p53 target genes from DoRothEA [24] as the p53 regulon (see **Materials and Methods**), we computed intron-based and exon-based TF activities along the time course of cellular response to DNA damage (**S3A Fig**). We found that, as expected, intron-based p53 activities display a higher correlation with p53 chromatin occupancies compared to exon-based p53 activities (**Fig 2E**). Importantly, both estimated activities are significantly higher than estimated activities using random regulons as controls (**S3B Fig**).

Given the better performance of intron-based method compared to exon-based method, we asked whether the improvement in this dataset also depends on target genes' mRNA stabilities, similar to the NF-κB dataset (**S2F Fig**). Analogous to **S2D Fig**, we created two sub-regulons of p53 with target genes having either short or long mRNA half-lives. Using these two sub-regulons, we reassuringly found that the improvement of intron-based method indeed depends on the mRNA stability for both replicates (**S3C Fig**). This result provided an additional line of support for our finding that mRNA stability can greatly affect TF activity estimation.

Using this dataset, we further explored how regulon choice could influence TF activity estimation. Because we so far have used cell-type non-specific TF regulons (from the DoRothEA database [24]), we asked whether refining the regulon using cell-type-specific TF binding data (such as ChIP-seq) would improve activity estimation. We thus used MCF7 p53 ChIP-seq data to create a refined p53 regulon (see **Materials and Methods**). We found that this refined regulon greatly increases the fold-changes of estimated TF activities for both intron-based and exon-based methods (compare **S3D Fig** with **S3A Fig**), and importantly, increases the correlation between estimated TF activities with p53 DNA binding activities (compare **S3D Fig** with **Fig 2E**). These results indicate that TF activity estimation can be greatly improved by using curated cell-type-specific regulon information.

Together, results from the p53 dataset further support that intron-level information provides a better estimate of the instantaneous TF activity. More importantly, with intron-based TF activity, we could accurately dissect the temporal activity dynamics of TFs such as p53 and NF-κB that control key biological processes. It should be noted that the overall performance increase for intron-based method compared to exon-based method is not as large as shown in the simulation (e.g., **Fig 1C**), which could be due to the sparsity in gene expression data (especially intronic expression counts, see **S1C and S1D Fig**) and the choice of the regulon.

## Intron-based TF activity recapitulates the temporal phasing of circadian TFs

Thus far, we have implemented a simple model describing transcription and splicing to illustrate the advantage of using intron-level information for capturing TF activity, and have used two public RNA-seq datasets to demonstrate that TF activity based on intronic read counts

can accurately capture the instantaneous TF activity. These results highlight the potential for using intron-based TF activity for dissecting TF activity dynamics during temporal biological processes. To explore this, we next focused on combing this approach with time-series transcriptome data to study the dynamic regulation of key biological processes.

We first focused on decoding TF activity dynamics during circadian rhythm in the mouse liver. The rationale for using circadian rhythm as a case study is two-fold. On the one hand, light-entrainment of the circadian clock produces rhythmic regulatory signals that are synchronized among cells for an extended time [10], allowing us to use bulk-level time-series transcriptome data to decode the TF activity dynamics. In contrast, without external entrainment, other TFs such as p53 in the preceding example would quickly lose synchrony across cells, and bulk-level time-series transcriptome data would not be able to capture long-term TF activity dynamics. On the other hand, time-series circadian TF ChIP-seq data [33] allows us to compare the estimated TF activity dynamics with the ChIP-based TF activity dynamics.

We resorted to a public time-series RNA-seq data of circadian-entrained mouse livers collected with a relatively high temporal resolution (i.e., every 2 hours) [42] (**Fig 3A**). Using this dataset and the curated TF regulons (**Materials and Methods**), we estimated TF activity dynamics using either intronic or exonic information, and found 13 TFs whose activity dynamics estimated from both measures are rhythmic (**S4A Fig** class 1). For the two classes

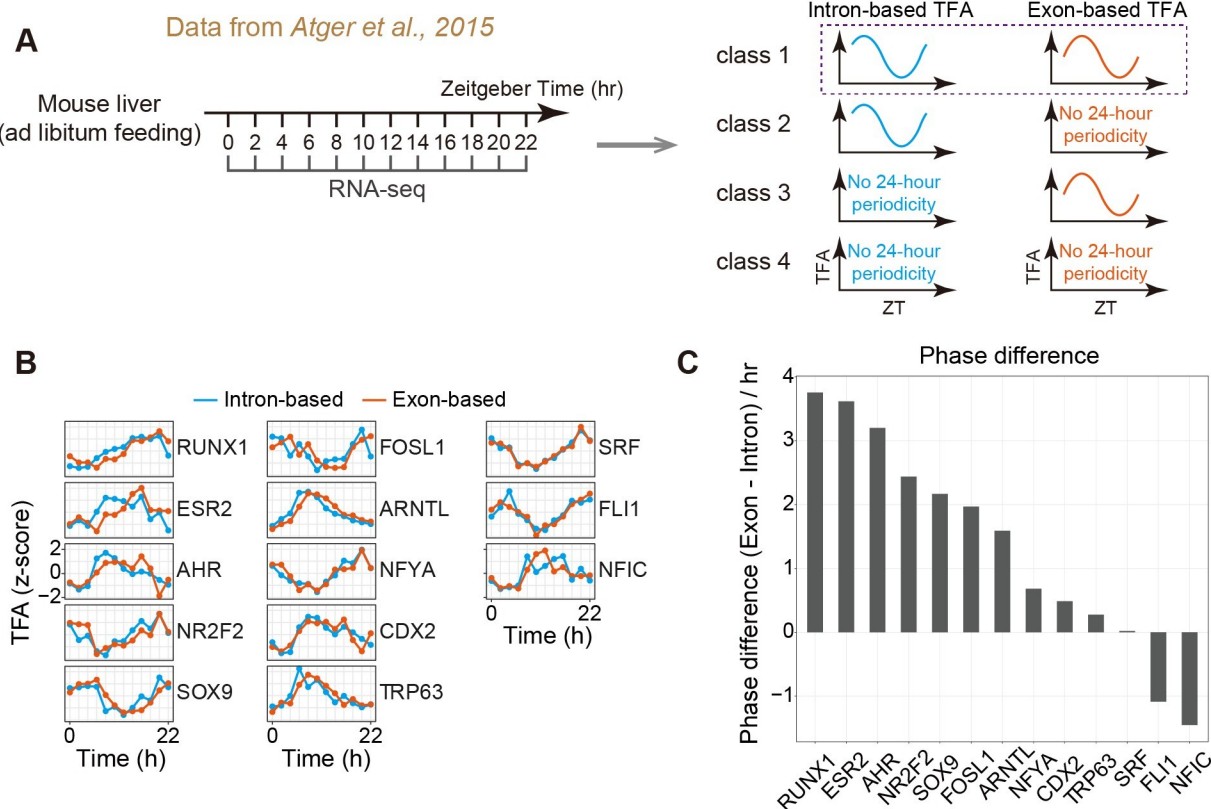

**Fig 3. Analysis of rhythmic TFs during mouse liver circadian rhythm. (A)** Schematic of the experimental design of Atger et al [42] and the approach to identify circadian TFs by using estimated TF activity (TFA). Time-series RNA-seq data were used to calculate TFA dynamics, and TFs with 24-h periodicity were identified as circadian TFs. To compare between intron-based and exon-based TFAs, we focused on class 1 TFs (see also **S4A Fig**). **(B)** TFA dynamics of 13 class 1 TFs. Intron-based TFA and exon-based TFA were z-score normalized. Note that all these regulons have confidence scores of A or B in the DoRothEA database. **(C)** Phase differences between exon-based TFA and intron-based TFA for class 1 TFs. Note that positive values indicate that exon-based TFA lags behind intron-based TFA.

where only one type of TF activity dynamics was identified to be rhythmic (i.e., class 2 and 3), the activity dynamics of example TFs (**S4B Fig**) indicate that the reason why one of dynamics was not rhythmic could be at least two-fold. First, the algorithm for detecting rhythmicity could be sensitive to the noise in the data. Second, there could be biologically meaningful difference between intron-based and exon-based activity dynamics, causing one to be rhythmic but not the other. Nevertheless, we focused on the class 1 TFs and compared the phasing of their periodic dynamics. Most TFs' (11 out of 13) intron-based activity dynamics display a forward-shifted circadian phasing compared to exon-based dynamics (**Fig 3B and 3C**), which is as expected. For the two TFs that display a backward-shifted phasing, we found that it is likely due to noise in the data, as the result is not robust to biological replication (**S4C Fig**). Thus, intron-based TF activity allows capturing rhythmic circadian TF dynamics.

The difference in phase difference between intron-based and exon-based activity dynamics for different TFs promoted us to investigate the potential mechanism. A possible explanation is that target genes of different TFs have different intron lengths, leading to different processing times of the target mRNAs and thus different phase differences. However, we found that phase difference does not correlate with target intron length (**S4C Fig**). Thus, other mechanisms involving RNA processing likely account for the difference in phase difference.

We next compared the circadian phasing of intron-based TF dynamics with the circadian phasing measured by ChIP-seq. We first focused on a well-known circadian TF, BAML1 or ARNTL, and found that intron-based BAML1 activity peaks between 6–8 hour (**Fig 3B**), which is close to the ChIP-seq result (i.e., 6.1 hour) [33]. In contrast, exon-based BAML1 activity peaks at ~ 8 hour (**Fig 3B**). To compare our results with more ChIP-seq dynamics of additional TFs, we loosened the stringency in the curated regulon (see **Materials and Methods**) and computed the activity dynamics of CLOCK, another key circadian TF. We found that intron-based CLOCK activity peaks at ~ 8 hour, close to the ChIP-seq result (i.e., 7.3 hour) [33], whereas exon-based CLOCK activity peaks at ~12–14 hour (**S4D Fig**).

The above results suggest that intron-based TF activity can accurately recapitulate rhythmic TF dynamics, which provides a useful tool for studying circadian transcriptional architecture, as many TFs do not have existing time-series ChIP-seq data (e.g., many TFs in **Fig 3B**).

## Global TF activity profiling revealed two temporally opposing TF modules that dictate T cell response

We next acquired and analyzed our own data on transient T cell responses to chemical stimulation. T cell activation is a complex dynamic process that depends on the phosphorylation cascade upon T cell receptor engagement, and involves many TFs, including NFAT, AP-1, and NF-κB [43]. Although target genes of these TFs have been identified, we still lack an overall understanding of transcriptional events during early T cell activation.

To address this, we performed time-series bulk RNA-seq of Jurkat T cells post the stimulation by PMA and ionomycin, a chemical mimic of TCR stimulation [44] (**Fig 4A**). As discussed above, cells would lose synchrony at later time points, and we thus collected the samples at relatively early time points post stimulation. More specifically, we collected samples at 8 time points within 1-hour post stimulation (**Fig 4A**). Using these data, we aimed to decode the transcriptional architecture during early T cell activation by profiling TF activities globally (see **Materials and Methods**).

To do so, we computed the correlation between intron-based TF activities for each pair of TFs and performed hierarchical clustering. Intriguingly, we identified two TF modules enriched for different functions and displaying opposing temporal profiles (**Fig 4B**). More specifically, the first module is enriched for TFs participating in stress responses, and displays an

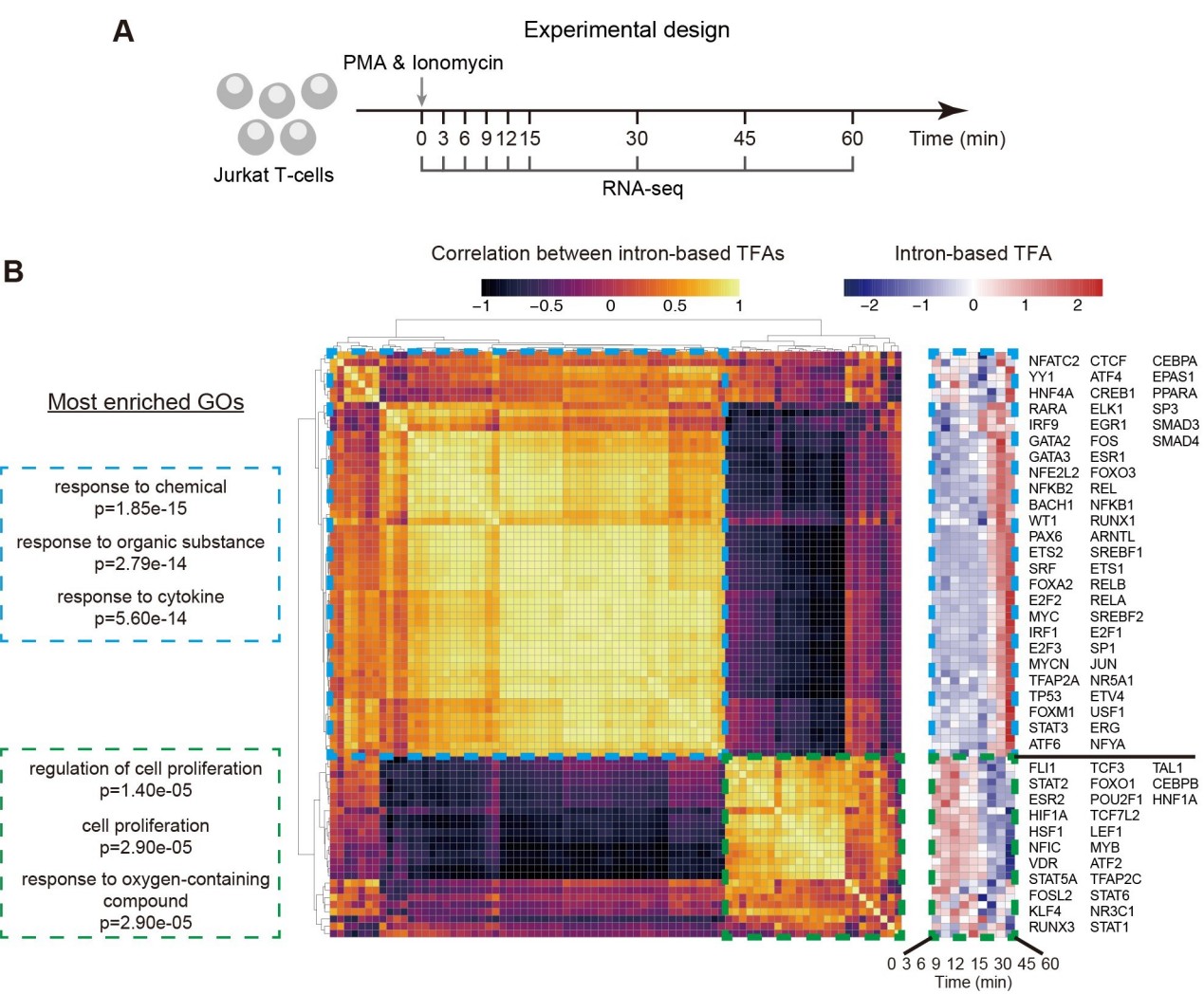

**Fig 4. Decoding global transcriptional regulation during the early activation of Jurkat T cells by profiling intron-based TF activities. (A)** Schematic of our experimental design. Time-series RNA-seq data were acquired during early Jurkat T cell activation. Each time point has two replicates. **(B)** Hierarchical clustering uncovered two functionally and dynamically distinct TF modules. Clustering was performed based on the Pearson correlation between intron-based TFAs. Two TF modules are indicated by dashed boxes. Three most enriched GO biological process terms for each module are shown. Intron-based TFA dynamics after z-score normalization of each TF are shown on the right. Note that only TFs with non-zero expression values were included in the analysis (Materials and Methods), and TF symbols are ordered from top to bottom, then left to right. See also **S5 Fig** for analogous analyses using exon-based and total reads-based methods.

increase in TF activity post stimulation, suggesting a gradual engagement of the stress response mechanism over a time course of ~ 1 hour. In contrast, the second module is enriched for TFs mediating homeostatic processes such as cell proliferation, and displays a decrease in TF activity post stimulation. Importantly, the same analysis performed with exon-based TF activity (**S5A Fig**) or total reads-based (i.e., intron plus exon reads) TF activity (**S5B Fig**) resulted in TF modules with overlapping functions, i.e., both modules in both methods are enriched for the GO term "response to chemical".

These results indicate a potential temporal organization program during early T cell response, in which the gene regulatory system globally and quickly transits from homeostasis to stress response within an hour after stress begins. While these results suggest that intron-based method can provide biologically important insights into dynamic T cell response, the

extent and function of the observed large-scale changes in TF activities necessitate further investigations.

We further examined individual TFs involved in T cell activation that displayed differential activity dynamics when estimated with different methods. The two notable TFs are IRF9 and GATA3, both are known to play key roles in immune response. With intron-based method, the two TFs displayed a two-pulse-like activation (**S6 Fig** top panels). In contrast, with both exon-based and total reads-based methods, they displayed a single pulse activation (**S6 Fig** middle and bottom panels). These results indicated that the latter two methods are comparable, and that the intron-based method might reveal intriguing TF dynamics that would be worth investigating further.

Thus, by performing time-series bulk RNA-seq of Jurkat T cells at short time points post stimulation, we demonstrated the application of using intron-based TF activity estimation for dissecting the temporal design principle of gene regulatory network.

## Discussion

Gene regulatory network is highly dynamic and TFs in the network can display complex temporal activity dynamics. While single-cell time-lapse imaging has been typically used for analyzing these dynamics, a genome-wide approach for analyzing multiple TFs in a high-throughput manner would be desirable. In this work, by combing computer simulation, analyses of public datasets, and the generation of our own datasets, we showed that intron-level information, together with literature curated regulon information, can allow decoding TF dynamics using time-series RNA-seq data. Consistent with previous studies that leverage intron-level information to analyze transcriptional dynamics [33, 34] and cell state transition [35, 36], our study demonstrates the power of using intronic read counts in typical RNA-seq datasets for understanding global transcriptional architecture. While experimental methods have been developed to specifically measure newly transcribed RNAs [28–30], which also allow accurate decoding of TF activity dynamics [31], our method can take advantage of the existing public RNA-seq datasets.

Three fundamental elements are important for applying our proposed approach for broadly analyzing TF activity dynamics. Firstly, the regulon-wide short-lived, unspliced mRNAs allow one to accurately read out upstream TF activity, which we demonstrated by simulation and by comparing with TF nuclear localization dynamics and TF DNA binding activity dynamics. However, because TF nuclear localization or TF DNA binding activity may not be an accurate surrogate of TF activity under certain scenarios, additional analyses may be necessary to further compare intron-based TF activity with the actual TF activity.

Secondly, cells in a population often maintain a high degree of synchrony for a short period of time after being subjected to a transient stress, allowing the analysis of TF dynamics within this short period using time series snapshots (of different cell populations). In other words, because of the synchrony, it is reasonable to assume that we are effectively measuring a meta-cell (representing the average of the population) by taking multiple snapshots within this short period. This concept could be better understood when considering the time-series tissue-level RNA-seq data during the circadian clock. In this scenario, it is typically assumed that all cells are synchronized throughout the circadian clock, as cells are entrained by an external light-dark clock, and we are thus following a meta-cell undergoing the circadian rhythm. However, cell population would lose synchrony quickly after step-wise stimulations, the timescale of which would need to be determined for each system of interest.

Thirdly, our method exploits the well-curated gene regulatory network, in which TF regulons have been carefully assembled for over a hundred TFs [24, 40]. However, there are two

related limitations. First, the curated regulon information is not cell-type specific, and different cell types may have different network wiring, presenting a potential compounding factor for accurate TF activity estimation. And as we have demonstrated using the p53 dataset, refining the regulon with cell-type-specific ChIP-seq data can increase the accuracy of estimated TF activities (**S3D and S3E Fig**). Second, a large fraction of TFs still does not have well-curated regulon information [24, 40], prohibiting an accurate activity analysis of these TFs. Thus, the ability to decode dynamic biological processes using intron-level TF activity would be further enhanced by the continuing understanding of gene regulation and gene regulatory networks.

While our method is built upon existing methods using exon-level information of target genes to estimate upstream TF activity [17–25], it provides some advantages compared to exon-based methods. A key advantage is that intron-based TF activity captures the instantaneous activity of the TF, and in contrast, exon-based TF activity captures the integrated activity of the TF. Thus, intron-based TF activity allows us to accurately decode the temporal regulation of TFs during dynamic biological processes. It should be noted that the difference between the two TF activity measures depends on the turnover rate of the matured mRNAs of target genes (as we have demonstrated in both simulations and experimental datasets), and thus for some TFs, there may not be an advantage for using the intron-based measure. Furthermore, some target genes do not have introns or may not have intronic reads, prohibiting the application of our method for TFs regulating these genes. Additionally, while our method has been demonstrated using both single-cell RNA-seq and bulk RNA-seq data, we noted that the low capturing rates of RNAs, especially unspliced pre-mRNAs, could significantly affect the performance of the method (**S1C and S1D Fig**)

Compared to analyzing TF activity dynamics using time-lapse imaging [4–6], intron-based TF activity cannot resolve dynamics of timescales shorter than splicing due to the theoretical limitation. Furthermore, it can only capture population-averaged dynamic behaviors, which acts as a low-pass filter to further decrease the temporal resolution of the approach. Despite these limits, intron-based TF activity offers a temporal picture of global transcriptional architecture during dynamic processes such as early T cell activation, allowing us to dissect the temporal organization principles of gene regulatory networks. We envision that this approach would be broadly applied to study diverse gene regulatory systems undergoing dynamic alterations.

## Materials and methods

### Simulation of the p53 system

Our simulations were mainly focused on the p53 system, in which p53 activity can exhibit oscillatory dynamics with relatively stable amplitude, duration, and period after DNA damage in single cells [12, 38, 39]. In the simulation, we assumed that p53 activity is maintained at a basal level before DNA damage and undergoes periodic oscillation after DNA damage. p53 dynamics, transcription, and splicing of target genes were modeled by the following ordinary differential equations:

$$\text{TF}(t) = A_{basal} \cdot \frac{1 + \cos(\frac{2\pi}{T}t)}{2} + A_{max} \cdot \frac{1 - \cos(\frac{2\pi}{T}t)}{2} + \text{noise} \tag{1}$$

$$\frac{du}{dt} = \alpha \frac{\text{TF}^n}{\text{TF}^n + K_d^{\ n}} - \beta u \tag{2}$$

$$\frac{ds}{dt} = \beta u - \gamma s, \tag{3}$$

where TF denotes p53 activity, u and s denote unspliced and spliced mRNA level of target gene respectively, t denotes time after DNA damage stress, $A_{basal}$ denotes the basal activity of p53, $A_{max}$ denotes the maximum activity of p53 during oscillation after DNA damage stress, T denotes the period of p53 oscillation, noise denotes a Gaussian noise with mean 0 and variance $\sigma^2$, α denotes the maximum capacity of target gene activation by p53, β denotes the splicing rate of target gene, n and $K_d$ denote Hill coefficient and the dissociation constant in the input function of p53 target gene respectively, and γ denotes degradation rate of the spliced mRNA of target gene.

Some of the parameter values are based on the literature. More specifically, $A_{basal}$ and $A_{max}$ are 0.06 μM and 0.5 μM respectively [45], and T is 5.5 h [38]. Since splicing typically completes within 5–10 minutes where 90% of unspliced mRNAs are spliced [46], we assumed that splicing follows exponential decay with β = log10/(splicing time), and splicing time was randomly chosen between 5 and 10 minutes for each target gene. Similarly, since previous study experimentally measured mRNA half-lives in MCF7 cell line (a widely used cell line for p53-related experiments) [47], thus γ = log2/(mRNA half life). Values of other parameters were chosen empirically, namely σ was set to be 0.2·$A_{basal}$, α was randomly chosen between 100 and 200 h$^{-1}$, n was set to be 2, and $K_d$ was randomly chosen between 0.05 and 0.25 μM. The list of p53 target genes was from DoRothEA database [24], only target genes that are activated by p53 with confidence score A or B in the database were considered.

For stochastic simulations, a custom code based on "τ-leap" method [48] was used, which simulated the dynamics of u and s based on Eqs (2 and 3) and p53 dynamics from Eq (1). Initial values of u and s were non-zero steady-state solutions of Eqs (2 and 3) with p53 maintained at the basal level. Duration and step size of simulation are 20 h and 0.001 h, respectively.

After stimulating unspliced and spliced mRNA levels of all target genes of p53, the average of unspliced (or spliced) mRNA levels of target genes was calculated as the estimated p53 activity, which was then compared with the actual p53 activity (namely input p53 activity in the simulation).

In order to investigate how sparsity of data affects the results, we performed simulations of a specific gene (with α = 100 h$^{-1}$ and a 100 min half-life) by subsampling unspliced and spliced mRNA counts with capture (i.e., detection) efficiency p. For both spliced and unspliced mRNAs, we performed different sets of simulations with p spanning 5%, 10%, 15%, 20%, 25% and 30% respectively.

## The general pipeline for TFA estimation using RNA-seq data

The first step in TFA calculation is to map the raw RNA-seq data in cases where intronic counts of genes are not provided. More specifically, raw RNA sequencing data was downloaded as fastq format using fastq-dump. After trimming and filtering with Trimmomatic version 0.38 [49], sequencing reads were aligned with STAR version 2.5.3a [50] to genome reference GRCh38.p13 for human or GRCm38.p6 for mouse from GENCODE [51]. Intron-level and exon-level read counts of each gene were calculated by TPMCalculator version 0.03 [52] for each sample and were normalized as counts per million (CPM) based on the following equations:

$$\text{CPM (intron) of gene A} = 1e6 \cdot \frac{\text{intron-level read count of gene A}}{\text{intron-level plus exon-level read count of all genes}} \qquad (4)$$

$$\text{CPM (exon) of gene A} = 1e6 \cdot \frac{\text{exon-level read count of gene A}}{\text{intron-level plus exon-level read count of all genes}} \qquad (5)$$

Note that the length of intronic or exonic regions was not used for normalization, since intronic reads might arise from priming at intronic-polyT regions [35] and thus the actual length of intronic regions that contribute to intronic reads is shorter than annotated length.

Next, TFAs were calculated using intron-level or exon-level CPM values of all genes in each sample. More specifically, for each TF in the sample, intron-level CPM values of its target genes were averaged as the intron-based TFA. Note that only genes with introns were included in the calculation. Meanwhile, exon-level CPM values of the target genes were averaged as the exon-based TFA. Lists of target genes (i.e., regulon information) of all TFs are from the DoRothEA database [24], and only target genes activated by the corresponding TF with confidence score A or B are included in the calculation unless otherwise specified.

## Analysis of public bulk RNA-seq data of mouse liver undergoing circadian rhythm

Time series mouse liver RNA-seq data during circadian clock were obtained from the study by Atger *et al.* [42]. The data for livers from C57BL/6J mice under ad libitum feeding were used for our analysis. In this public dataset, sequencing was performed every 2 h from ZT0 (Zeitgeber Time 0) to ZT22 with four biological repeats. Note that the calculation of intronic and exonic expression levels was performed in the original study, and intron-level and exon-level RPKM (Reads Per Kilobase per Million mapped reads) values of all genes were provided by the original study. We then averaged expression levels from four biological replicates in each time point and computed intron-based and exon-based TFAs along the circadian clock.

To search for TFs displaying circadian rhythm, MetaCycle version 1.2.0 [53] was used to determine whether intron-based or exon-based TFA exhibits 24-hour periodicity. More specifically, the period was fixed at 24 h and TFAs whose p values are lower than 0.05 (i.e., meta2d_pvalue<0.05) were defined as exhibiting circadian rhythm. The phases of TFAs of circadian TFs were computed by MetaCycle (i.e., meta2d_phase).

## Analysis of public NF-κB data

Multimodal data containing single-cell RNA-seq data and time-lapse microscopy data of p65 nuclear localization in the same single cell were obtained from the study of Lane *et al.* [41]. In their study, RAW 264.7 cells were stimulated with LPS, and time-lapse microscopy of p65-Clover was conducted in each cell, and after imaging the same cell was sequenced to obtain the transcriptome.

Intron-based and exon-based TFA of NF-κB in each cell were computed from single-cell RNA-seq data. Nuclear localization level of NF-κB was directly obtained from the original study and the data of the last frame was utilized. Cell numbers were 145, 365, and 124 for 75 min, 150 min, and 300 min after stimulation respectively.

We compared our method based on averaging with other methods for computing TF activities, namely AUCell [22] and VIPER [23]. AUCell is an algorithm to score the activity of each regulon in each cell. AUCell calculates the enrichment of the regulon as an area under the recovery curve (AUC) across the ranking of all genes in a particular cell, whereby genes are ranked by their expression value [22]. VIPER (virtual inference of protein activity by enriched regulon analysis) is an algorithm to estimate protein activity from gene expression data. From expression profile data, gene expression signatures (GES) are computed. Incorporating the regulon information, GES is then transformed into protein activity profile by aREA algorithm [23].

In order to investigate the effect of mRNA half-life on TF activity dynamics, we divided the NF-κB target genes into two sub-regulons according to their mRNA half-lives [47], calculated

the TFA for the two sub-regulons, and compared them with nuclear localization level of NF-κB. We analyzed the effect of mRNA half-life for both intron-based TFA and exon-based TFA. We also analyzed the effect of mRNA half-life on individual genes. For each gene, we calculated the mean expression (intron level or exon level) for all single cells at 4 time points. Then we calculated the correlation of mean expression with nuclear localization level of NF-κB. Using this correlation as a metric, we compared intron level with exon level, and plotted the intron correlation versus exon correlation ratio with mRNA half-life for each gene.

## Analysis of public p53 data

p53-related RNA-seq and ChIP-seq data were obtained from the study of Hafner *et al.* [39]. In their study, MCF7 cells were treated with γ-irradiation to activate oscillatory dynamics of p53 and p53 ChIP-seq was conducted at 0, 1, 2.5, 4, 5, and 7.5 h after stimulation without biological repeats, while RNA-seq was conducted every hour from 0 to 12 h after stimulation as well as 24 h after stimulation with two biological replicates.

We then computed cumulative p53 ChIP-seq signals in the whole genome as the genome-wide DNA binding activity, which was used as the surrogate of p53 activity. Meanwhile, RNA-seq data was analyzed to obtain intron-based and exon-based TFAs of p53. Since the sampling time was different between ChIP-seq and RNA-seq experiments, linear interpolation was performed before computing the correlation between the dynamics of the DNA binding activity from ChIP-seq and the estimated TFA of p53. The time points after interpolation were spaced by 0.5 h from 0 to 7.5 h.

We also analyzed the effect of mRNA half-life on estimated TFA. Using mRNA half-life data [47], we divided the genes into two sub-regulons, and compared the results of intron-based TFA and exon-based TFA.

To investigate how the choice of regulon affects our results, we used random regulons as negative control. We extracted all targets from DoRothEA database, excluded the targets of p53, and then subsampled this gene set 1000 times to obtain random regulons and calculate control TF activities for both intron-based and exon-based methods.

We also investigated how the cell-type-specific regulon affects the results. Using MCF7 ChIP-seq signals [39], we extracted p53 target genes whose TSS is within +/- 2kb of ChIP-seq peaks (195 targets). We used those p53 targets to refine the p53 targets from the DoRothEA database (i.e., overlapping subset), and calculated TFA with this refined regulon (42 target genes).

## RNA-seq experiment during Jurkat T cell activation

Jurkat T cells (ATCC, Clone E6-1) were cultivated in RPMI-1640 Medium (Gibco, catalog number C11875500BT), supplemented with 10% fetal bovine serum (Gemcell, catalog number 100–500) and 1% penicillin-streptomycin (Gibco, catalog number 15140–122). Cells were cultured at a concentration between $5x10^5$ and $5x10^6$ cells/mL and incubated under 37˚C and 5% $CO_2$.

For time series RNA-seq experiments, cells were plated at a density of $5x10^6$ cells/mL in 1 mL media and rested for 30 min before stimulation. Cells were then stimulated with 50 ng/mL PMA (Sigma-Aldrich, catalog number P1585) and 1 uM Ionomycin (Sigma-Aldrich, catalog number 407950) for different durations, namely 3, 6, 9, 12, 15, 30, 45, and 60 min. Note that cells without stimulation were also collected at 0 min. At each time point of sample collection, cells were centrifuged at 300 g for 1 min, and the supernatant was replaced by 1 mL Trizol (Invitrogen, catalog number 15596026). Gently mixed cell lysate was then flash-frozen in liquid nitrogen and preserved under -80˚C. MagZol Reagent Kit (Magen, catalog number R4801)

and VAHTS mRNA-seq V3 Library Prep Kit for Illumina (Vazyme, catalog number NR611) were used for RNA extraction and library construction. Sequencing was performed on an Illumina NovaSeq 6000 instrument in PE150 mode. Both library construction and sequencing were performed by GeneWiz.

For the analysis of Jurkat RNA-seq data, intron-based and exon-based TFAs of each sample were calculated and results of two biological replicates were averaged. One replicate for the 15 min sample showed abnormal TFA profile and was thus excluded from further analyses. Only TFs with non-zero expression values (namely, intron-level CPM plus exon-level CPM was greater than 0) were considered. GO analysis of TF modules was performed with R package *clusterProfiler* [54]. Only GO terms of biological process were considered and background genes for GO analysis were all TFs in human cells based on *RcisTarget* package [22]. Adjusted p-values based on Benjamini-Hochberg method were shown.

## Supporting information

**S1 Fig. Simulated results of the p53 system and the method for TFA calculation. (A)** Representative simulated traces of p53 dynamics, unspliced mRNA dynamics and spliced mRNA dynamics of three example genes. The spliced mRNA dynamics of the three genes are different due to their different mRNA half-lives. **(B)** The pipeline for calculating TFA using target expression levels from the simulation. The average expression level of target genes in the regulon of a TF is used as the estimate of the TF's activity. Either unspliced or spliced mRNA expression level can be used in the calculation. **(C-D)** The effect of detection rate (i.e., mRNA capture rate) on the correlation of gene expression level with input TFA. A p53 target gene was simulated at varying detection rates (such as 10% in **C**, see Materials and Methods), and the correlation between unspliced or spliced mRNA and input TFA was shown (**C**). The ratio between the two correlations were calculated for all simulated pairs of detection rates (**D**). (TIF)

**S2 Fig. Additional characterizations of the NF-κB dataset. (A)** Population-averaged NF-κB nuclear localization dynamics of the data from Lane et al. n = 637 cells for t = 0–75 min, n = 492 cells for t = 80–150 min, and n = 124 cells for t = 155–300 min. Error bar represents the standard deviation. **(B)** Analysis of individual NF-κB target genes. We focused on high-expressing genes (exonCPM >1 & intronCPM >1). In 39 high-expressing target genes, 26 genes showed positive correlation for both intronCPM and exonCPM. See Materials and Methods for details. **(C)** The effect of mRNA half-life on the relative performance of intron-based method versus exon-based method. For the 26 genes with double-positive correlation in (B), 15 of them have mRNA half-life data. The ratio between the intron-based correlation and exon-based correlation was plotted against mRNA half-life. Pearson correlation (R) and the associated p-value were indicated. **(D)** Schematic showing the division of NF-κB regulon into two sub-regulons. **(E-F)** Comparison of using the two sub-regulons for TFA estimations. Scatter plots showing TFAs estimated using the two sub-regulations versus NF-κB (p65) nuclear localization level in individual cells at 75 min post LPS stimulation **(E)**. The correlations between estimated TFAs and TF nuclear localization were shown for both sub-regulons **(F)**. (TIF)

**S3 Fig. Additional characterizations of the p53 dataset. (A)** Dynamics of p53 DNA binding activity, intron-based TFA, exon-based TFA and p53 expression (averages of two replicates). These dynamics were analyzed from the data of Hafner et al. **(B)** The comparison between estimated TFAs and control TFAs. The distribution of the control TFAs (dotted lines) were calculated from 1000 random regulons sampled from non-p53 target genes (see Materials and

Methods). The vertical lines indicate the estimated TFAs using the actual p53 regulon. **(C)** Comparison of using the two sub-regulons containing p53 target genes with either long or short mRNA half-lives for TFA estimations. The sub-regulons were generated as in **S2D Fig**. See Materials and Methods for details. **(D-E)** p53 activity estimations using the refined p53 regulon. The cell-type-refined p53 regulon was obtained as described in the Materials and Methods. This refined regulon was then used to generate plots analogous to panel A (**D**) and **Fig 2E** (**E**)
(TIF)

**S4 Fig. Additional analyses of circadian TFs. (A)** Classification of TFs based on the periodicity of TFA. P-values of the periodicity were calculated using intron-based and exon-based TFAs. Black points represent non-circadian TFs by both TFAs. Red points represent circadian TFs identified by intron-based TFA. Blue points represent circadian TFs identified by exon-based TFA. Purple points represent circadian TFs identified by both methods, which were used for the analysis in **Fig 3**. **(B)** Example TFA dynamics of TFs from class 2 and class 3. Class 2 TFs were taken from the red points in panel **A**, while class 3 TFs were taken from the blue points. **(C)** Robustness of circadian TFs to experimental replication. By down-sampling of 4 replicates, the robustness of each circadian TF was defined as the fraction of attempts that the TF was identified as circadian TF. **(D)** Target intron lengths of TFs with different phase differences. **(E)** Intron-based versus exon-based TFA dynamics of CLOCK protein. Note that the regulon of CLOCK is only available when loosening the confidence threshold of the regulatory links to the third grade (i.e., score C) in the DoRothEA database.
(TIF)

**S5 Fig. Hierarchical clustering analyses for exon-based TFA (A) and total reads-based TFA (B).** These analyses are analogous to **Fig 4B**. Two TF modules in each panel are indicated by dashed boxes. Three most enriched GO biological process terms for each module are shown. The GO highlighted in purple indicates a shared GO between two modules in each panel. Exon-based TFA dynamics and total reads-based (exon plus intron) TFA dynamics after z-score normalization of each TF are shown on the right. Note that TF symbols are ordered from top to bottom, then left to right.
(TIF)

**S6 Fig. Estimated activity dynamics of two example immune-related TFs during the response of Jurkat T cells to chemical stimulation.** For each TF, the activity dynamics estimated by three different methods were calculated using the time-series transcriptome data (**Fig 4**). These two TFs were chosen as examples that the intron-based method can yield drastically different activity dynamics as compared to the other two methods. Note that the exon-based and total reads-based methods generally produce very similar activity dynamics.
(TIF)

**S1 Table. Detailed information of public datasets used in this study.**
(XLSX)

**S2 Table. Source data for figures.**
(XLSX)

## Acknowledgments

The authors would like to thank the High Performance Computing Platform of the Peking-Tsinghua Center for Life Sciences for computational support.

## Author Contributions

**Conceptualization:** Yan Wu, Yihan Lin.

**Funding acquisition:** Yihan Lin.

**Investigation:** Yan Wu, Lingfeng Xue, Wen Huang.

**Methodology:** Yan Wu, Wen Huang.

**Resources:** Wen Huang.

**Software:** Yan Wu, Lingfeng Xue.

**Supervision:** Minghua Deng, Yihan Lin.

**Validation:** Lingfeng Xue.

**Writing – original draft:** Yan Wu, Lingfeng Xue, Yihan Lin.

**Writing – review & editing:** Yan Wu, Lingfeng Xue, Wen Huang, Minghua Deng, Yihan Lin.

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
