## [Decision Letter · Decision Letter 0]

16 Sep 2021

Dear Dr. Lin,

Thank you very much for submitting your manuscript "Profiling transcription factor activity dynamics using intronic reads in time-series transcriptome data" for consideration at PLOS Computational Biology.

As with all papers reviewed by the journal, your manuscript was reviewed by members of the editorial board and by several independent reviewers. In light of the reviews (below this email), we would like to invite the resubmission of a significantly-revised version that takes into account the reviewers' comments.

We cannot make any decision about publication until we have seen the revised manuscript and your response to the reviewers' comments. Your revised manuscript is also likely to be sent to reviewers for further evaluation.

Sincerely,

Jing Chen

Guest Editor

PLOS Computational Biology

Ilya Ioshikhes

Deputy Editor

PLOS Computational Biology

Reviewer's Responses to Questions

**Comments to the Authors:**

Reviewer #1: In this manuscript, Wu et al. systematically compared the intronic reads and exonic reads for TF activity inference analysis in both simulated and real time-series RNA-seq data. Sufficient evidence is provided to support their main conclusion that the intron-level information better recapitulates TF activities. They further applied this strategy to identify main TF modules regulating T cell activation in a newly generated dataset. I am generally satisfied with the main analysis and the quality of the new dataset included in the manuscript. Following are comments that should be fixed before the publication of this manuscript.

1. The intronic-information significantly outperforms the spliced mRNA in the simulated data (fig. 1), but the different is not so dramatic in the real dataset. The specific reasons should be discussed and clarified in the manuscript.

2. One potential limitation of this strategy is that the intronic signal is only a small subset of the gene expression information. How does the data sparsity of the intronic information affect the analysis? Does the same strategy works for single cell RNA-seq analysis?

3. For comparison analysis, it would be helpful to also include normal gene quantification (exon + intron) as this is normally used in conventional TF inference analysis.

4. Since the full gene expression or exonic reads can be used to infer the TF modules involved in T cell activation, what is the unique biology revealed by the intronic-level analysis? This should be clarified in the manuscript.

Reviewer #2: In this article, Lin and colleagues report a method to more accurately infer transcription factor activity (TFA) from RNA-seq data, where they use the intronic reads (as opposed to exonic reads) of transcription factor target genes (regulons) as measure of transcription factor activity. They show that TFA calculated using intronic reads correlate more strongly with transcription factor activity from simulated data (Fig. 1), single-cell nuclear TF translocation imaging measurements (Fig. 2A-C), TF binding data for both p53 (Fig. 2D) and for the circadian TFs ARNTL and CLOCK (FIg. 3 and Fig. S3D). Finally, they show that TFA calculated from intronic reads show partitioning into two distinct modules upon T cell activation (Fig. 4).

I found this work manuscript to be clearly written, and the for the most part, the analysis was clearly described and soundly executed. However, I found the improvements in TFA estimation using intron reads to be only marginally better compared to those obtained using the exon reads (e.g. slightly better correlations with TF binding data), and was not convinced this method was a substantial improvement that could yield new insights when applied to existing transcriptome data, which was touted as a strength of this approach. Analysis of TF activity from single-cell RNA-seq in T cell activation (Fig. 4) yielded two transcription factor modules, which appears to be a new insight; however, as analysis of exon data was not shown, it is not clear whether analysis of exon reads alone, using published approaches, would have been sufficient to unveil these two TF modules.

Comments:

- Fig. 1: It needs to be mentioned in the figure and also the legend that these p53 TF activities and correlations are obtained from simulation data; otherwise, these results can easily be mistook as those generated from experimental data.

- Fig. 2: It was a really nice idea for the authors to use the data from Lane et al. for Fig. 2A-C, as this data set uniquely measures NFkB localization and target gene expression in the same single cell. I feel that the authors should make use of this data by showing intron transcript levels and nuclear localization dynamics for single cells, and at the level of individual genes. It would also be nice to show here (and elsewhere) that analyzing intronic reads better recapitulates NkB activity on long half-life transcripts versus on short half-life transcripts.

- Fig. 3 and S3. Why are some genes predicted to be oscillatory when analyzing introns or exon (Class 2 and 3)? The authors point out this category, but do not elaborate; it would be interesting to know why.

- Fig. 4: How are the regulons for the shown TFs obtained? It is not stated here or in the methods. Also, is there any evidence that the levels/activity of these TFs change over time? While I can imagine that there be activity changes for the TFs downstream of T cell receptor signaling (e.g. NFAT, AP-1, and NFkB), it seems implausible that the activities of all these TFs could be changing within such a short time frame of 60 minutes. This result calls into question the process as to how TF regulons are defined.

- Fig. 4: How do these TF modules obtained here compared to those obtained by analysis of exonic reads? If the authors are arguing that analysis of intronic reads better reveals these modules compared to analysis of exonic reads, they need to show the improvement to provide evidence that this technique is uniquely capable or at least more capable of generating biological insights.

Reviewer #3: Using publicly available data, the authors presented evidence supporting the concept that transcription factor (TF) activities inferred from intronic reads (used as a proxy for pre-mRNA level) of the TF targeted genes can better recapitulate instantaneous TF activities compared to the exon-based reads in time-series RNA-seq data. More specifically, they showed that intron-based inferred NF-kappaB activities are better correlated with the measured NF-kappaB activities at nuclear localization level in a time-series single-cell RNA-seq data. Subsequently, they showed that intron-based TF activities improve the characterization of the temporal phasing of cycling TFs during circadian rhythm. They further applied the approach to their own data on transient T-cell responses to chemical stimulation and revealed two temporally opposing TF modules during T cell activation using Jurkat T cells. This paper proposed a simple and but effective analysis strategy to infer dynamic TF activities from time-series RNA-seq data. The idea is presented clearly in the manuscript. However, I have the following questions on specific technical aspects.

Major comments

1. In Fig. 2C, the authors compared VIPER, AUCell results to demonstrate the robustness of the proposed method. It was unclear how VIPER, and AUCell compute the correlation. Was it because the target genes were different? How is this related to robustness? Some explanation is necessary.

2. It is interesting to explore the relationship between the TF chromatin occupancy from the time-series TF ChIP-seq data and TF activities inferred from the time-series RNA-seq data. As the authors also mentioned in Discussion, the curated TF target genes are not cell-type specific. Thus, when examining the p53 activity, the authors could refine the target genes using the TF ChIP-seq data, i.e., focus on those target genes with associated ChIP-seq peak signals. I wonder what impact it would bring on the results in Fig.2.

3. The estimated TFA for a TF is dependent on the target gene set from a database and in this study, the DoRothEA database was used. Especially in the paper, the genes with confidence score A or B for a TF are selected for the TFA estimation. As a negative control, it would be good to know how different that would impact the TFAs if random target genes are selected.

In addition, was mRNA-level of the TF itself considered? If a TF is expressed at a low level, do you still consider that the TF has a high activity level if the estimated TFA is a high value? Is there some conditioning on the TF mRNAs required? It was not clear from the paper.

Minor comments

1. The references sited seem to be inaccurate for several datasets used in the analysis.

Line2-4-Line 304: “We resorted to a public time-series RNA-seq data of circadian-entrained mouse livers collected with a relatively high temporal resolution (i.e., every 2 hours) [44] (Fig. 3A).”

However, this paper does not have the mentioned datasets. They are from ref.43?

2. In the Fig.3 legend, class 1 and class I are used. Keep it consistent.

3. Provide GEO numbers to the public datasets used in the analysis.

**Have the authors made all data and (if applicable) computational code underlying the findings in their manuscript fully available?**

Reviewer #1: Yes

Reviewer #2: Yes

Reviewer #3: **No: **On GitHub, the authors made code available, but they did not provide the datasets used in the analysis.

PLOS authors have the option to publish the peer review history of their article (what does this mean?). If published, this will include your full peer review and any attached files.

Reviewer #1: **Yes: **Junyue Cao

Reviewer #2: No

Reviewer #3: No
---

## [Decision Letter · Decision Letter 1]

15 Dec 2021

Dear Dr. Lin,

We are pleased to inform you that your manuscript 'Profiling transcription factor activity dynamics using intronic reads in time-series transcriptome data' has been provisionally accepted for publication in PLOS Computational Biology.

Best regards,

Jing Chen

Guest Editor

PLOS Computational Biology

Ilya Ioshikhes

Deputy Editor

PLOS Computational Biology

Reviewer's Responses to Questions

**Comments to the Authors:**

Reviewer #1: The authors have satisfactorily answered all my comments. I do not have further questions.

Reviewer #2: The authors have addressed my concerns, and I support publication of this manuscript.

Reviewer #3: The authors have addressed my questions and suggestions comprehensively. I have no futher comments.

**Have the authors made all data and (if applicable) computational code underlying the findings in their manuscript fully available?**

Reviewer #1: Yes

Reviewer #2: Yes

Reviewer #3: Yes

PLOS authors have the option to publish the peer review history of their article (what does this mean?). If published, this will include your full peer review and any attached files.

Reviewer #1: **Yes: **Junyue Cao

Reviewer #2: No

Reviewer #3: No

---

## [Editor Report · Acceptance letter]

5 Jan 2022

PCOMPBIOL-D-21-01294R1 

Profiling transcription factor activity dynamics using intronic reads in time-series transcriptome data

Dear Dr Lin,

I am pleased to inform you that your manuscript has been formally accepted for publication in PLOS Computational Biology. Your manuscript is now with our production department and you will be notified of the publication date in due course.

With kind regards,

Agnes Pap
